# A weighted region-based level set method for image segmentation with intensity inhomogeneity

**Haiping Yu**[1]*, **Ping Sun**[1], **Fazhi He**[2], **Zhihua Hu**[1]

**1** School of Computer Science, Huanggang Normal University, Huanggang, China, **2** School of Computer Science, Wuhan University, Wuhan, China

* seapingyu@outlook.com

## Abstract

Image segmentation is a fundamental task in image processing and is still a challenging problem when processing images with high noise, low resolution and intensity inhomogeneity. In this paper, a weighted region-based level set method, which is based on the techniques of local statistical theory, level set theory and curve evolution, is proposed. Specifically, a new weighted pressure force function (WPF) is first presented to flexibly drive the closed contour to shrink or expand outside and inside of the object. Second, a faster and smoother regularization term is added to ensure the stability of the curve evolution and that there is no need for initialization in curve evolution. Third, the WPF is integrated into the region-based level set framework to accelerate the speed of the curve evolution and improve the accuracy of image segmentation. Experimental results on medical and natural images demonstrate that the proposed segmentation model is more efficient and robust to noise than other state-of-the-art models.

## Introduction

Image segmentation has always been one of the difficulties in image processing and computer vision. The process of image segmentation divides an observed image into separate nonoverlapping regions in the image domain. Many methods have been proposed and have achieved many improvements. However, due to the influence of image equipment, noise and uneven lighting conditions, angiographic images, such as medical images, have the characteristics of intensity inhomogeneity, low contrast, blur and weak boundaries. Image segmentation always remains a challenging problem in the field of image processing. For image segmentation, first, processing was manually implemented by medical specialists, which was time-consuming and laborious. In addition, manual segmentation becomes impossible when the number of images is large or the number of images to be segmented increases. Later, many automatic segmentation methods were presented for image segmentation [1–5].

In recent years, deep learning, such as convolutional neural networks (CNNs) are widely used to segment images and is have achieved excellent results in the medical image segmentation. However, these methods require considerable training dataset and have a complex

**Data Availability Statement:** All relevant data are within the manuscript and its Supporting Information zip files.

**Funding:** This study was supported by the Hubei Provincial Natural Science Foundation of China in the form of a grant to HY (No. 2019CFB797) and

the Natural Key Research and Development Project of China in the form of a grant to FH (No. 2016YFC0106305).

structure and low execution efficiency. Although some deep segmentation models that support small sample data have recently appeared, these models often have over-segmentation or cross-learning problem. Therefore, Therefore, model-based image segmentation methods still have space for research, and these methods have higher time and space efficiency.

Due to the limitations of the present methods, significant improvements are still necessary. In recent years, the active contour model has been capable of providing closed contours of target objects and has been extensively applied to image segmentation and based on the theory of curve evolution and fluid mechanics [6–11]. In the active contour model, the level set method proposed by Osher and Sethian, provides a very effective framework for the numerical description of low-dimensional curves and high-dimensional surfaces [12], and is widely used in image processing and computer vision [13–16].

Level set methods implicitly represent geometrical objects such as curves or surfaces in terms of a level set function (LSF) and successfully capture topological changes of objects. In level set methods, researchers have proposed many classic segmentation models, including edge-based models (EBMs) and region-based models (RBMs). EBMs generally utilize a gradient consisting of a stop function shifting into an energy functional to segment object boundaries. The GAC model [17], a popular edge-based model, utilizes intrinsic geometric measures to make an edge stopping function (ESF) to evolve the curve. DRLSE [18] is another standard edge-based model that utilizes the external energy term to guide the closed curve towards the boundaries of the image. Subsequently, some researchers have proposed other edge-based segmentation models [19–21]. These methods have promoted the development of segmentation technology and made some contributions in dealing with image segmentation. However, the gradient stop function is sensitive to initial contour information and noise. Region-based models are superior to edge-based models. Region-based models are less sensitive to noise and initial contours. The CV model [22], proposed by Chan and Vese, is a popular region-based models that utilizes global regional information to segment images with intensity homogeneity. Region Scalable Fitting (RSF) [23], proposed by Li et al., is another typical RBM that uses local intensity mean information to accurately segment images with intensity inhomogeneity. However, the model is sensitive to the initial information and easily falls into the local optimal solution. In addition, the model cannot accurately estimate the bias field. Recently, local region-based models have been proposed to segment images with intensity inhomogeneity. Liu et al. considered the local image characteristics and proposed the local region-based Chan-Vese (LRCV) model [24]. Wang et al. combined local information with global intensity information to construct a LCV model [25]. The local intensity clustering (LIC) model defined a local clustering criterion function and converted it to a level set framework to obtain level set evolution. The local region fitting information (LRFI) model [26], proposed by Liu et al., utilized a controllable velocity coefficient to accelerate the curve convergence speed.

Even though these methods have achieved good performance in some practical applications, the existing methods have some shortcomings, including the following:

1. boundary disconnection at crossover regions;

2. poor segmentation in the dark regions of the vessel image;

3. inaccurate segmentation of vessel images with intensity inhomogeneity.

In this paper, we propose a weighted region-based level set method to segment images with intensity inhomogeneity. In the model, first, a new weighted pressure force (WPF) is proposed to adaptively modulate the contractility of the balloon forces inside and outside of the closed contour by restricting the unevenness of the intensity mean value inside and outside the closed curve; second, a faster and smoother regularization term is utilized to ensure the stability of

the curve evolution during the process of solving; finally, the WPF is integrated into the level set framework to improve the accuracy of segmentation images with intensity inhomogeneity.

## Previous works

### CV model

The CV model, as a classic typical energy-based segmentation model, is proposed to segment two-phase images, based on the assumption that the target is segmented and the background is intensity inhomogeneous. Let an image I(x) on the image domain ϕ, the energy function is defined as:

$$E(\phi) = \lambda_1 \int_\Omega |I(x) - c_1|^2 dx + \lambda_2 \int_\Omega |I(x) - c_2|^2 dx + \mu \int_\Omega \delta(\phi)|\nabla\phi|dx + \nu \int_\Omega H(\phi)dx \qquad (1)$$

where $\lambda_1$, $\lambda_2$, $\mu$ and $\nu$ are positive coefficients; $c_1$ and $c_2$ represent the mean intensities inside and outside of the closed curve C, respectively; $H(\cdot)$ is a Heaviside function and $\delta(\cdot)$ is the derivative of $H(\cdot)$.

$$\begin{cases} H_\varepsilon(x) = \dfrac{1}{2}\left(1 + \dfrac{2}{\pi}\arctan\left(\dfrac{x}{\varepsilon}\right)\right) \\[4mm] \delta_\varepsilon(x) = \dfrac{\partial H(x)}{\partial x} = \dfrac{1}{\pi} \cdot \dfrac{\varepsilon}{\varepsilon^2 + x^2} \end{cases} \qquad (2)$$

According to the variational method, the evolution equation of the curve is expressed as follows:

$$\frac{\partial(x,t)}{\partial t} = \delta(\phi)\left[\lambda_2(I(x) - c_2)^2 - \lambda_1(I(x) - c_1)^2 + \mu\, div\frac{\nabla\phi}{|\nabla\phi|} - \nu\right] \qquad (3)$$

The CV model can segment images with intensity inhomogeneity; however, it has more sensitivity to initialization information and low efficiency while segmenting images with high noise and severe intensity inhomogeneity.

### DRLSE model

The DRLSE model, proposed by Li et al. eliminates the need for reinitialization and ensures the correctness of numerical calculations. The energy function of the DRLSE model can be expressed as:

$$\begin{aligned} E(\phi) &= \mu R_p(\phi) + \lambda L_g(\phi) + \alpha A_g(\phi) \\ &= \mu \int_\Omega p(|\nabla\phi|)d\mathbf{x} + \lambda \int_\Omega g\delta(\phi)|\nabla\phi|d\mathbf{x} + \alpha \int_\Omega gH(-\phi)d\mathbf{x} \end{aligned} \qquad (4)$$

In Eq (4), the first item $R_p(\phi)$ is the level set regularization term to ensure the stability of the level set evolution. $L_g(\phi)$ denotes the length of the penalty term, which was first proposed by the geodesic active contour (GAC) model. $A_g(\phi)$ is a weighted area of the closed contour that is used to accelerate the motion of the curve evolution. $L_g(\phi)$ and $A_g(\phi)$ are called the external energy. $\mu$, $\lambda$ and $\alpha$ are positive constant coefficients. $g$ is the edge stop function (ESF). The

function of $p$ is the potential function. The definitions are as follows:

$$g = \frac{1}{1 + |\nabla G_\sigma * I|^2} \tag{5}$$

$$p(|\nabla\phi|) = \begin{cases} \frac{1}{(2\pi)^2}(1 - \cos(2\pi|\nabla\phi|)), if\,|\nabla\phi| \leq 1 \\ \frac{1}{2}(|\nabla\phi| - 1)^2, \ if\,|\nabla\phi| \geq 1 \end{cases} \tag{6}$$

In the above equation, $G_\sigma$ is a Gaussian kernel with the deviation $\sigma$ and the function $p$ is used to avoid reinitialization of the LSF.

The corresponding level set evolution equation of the DRLSE model is:

$$\frac{\partial\phi}{\partial t} = \mu div(d_p(|\nabla\phi|)\nabla\phi) + \alpha g\delta_\varepsilon(\phi) + \lambda\delta_\varepsilon(\phi)div\left(g\frac{\nabla\phi}{|\nabla\phi|}\right) \tag{7}$$

where $\mu$, $\alpha$ and $\lambda$ are constant parameters, and $d_p(|\nabla\phi|)$ is a smooth bounded function that can control the properties of forward and backward diffusion. $\delta$ is the Dirac delta function defined as:

$$\delta_\varepsilon(x) = \begin{cases} \frac{1}{2\varepsilon}\left[1 + \cos\left(\frac{\pi x}{\varepsilon}\right)\right], |x| \leq \varepsilon \\ 0, \qquad\qquad\quad |x| > \varepsilon \end{cases} \tag{8}$$

## LVC model

Zhang et al. proposed an active contour model (ACM) that was implemented with the selective binary and Gaussian filtering regularized level set method [27]. In the model, a new region-based signed pressure force (SPF) function was proposed to control the direction of curve evolution. Therefore, the model can shrink or expand adaptively. The SPF function is as follows:

$$spf(I(x)) = \frac{I(x) - (c_1 + c_2)/2}{\max(|I(x) - (c_1 + c_2)/2|)}, x \in \Omega \tag{9}$$

where $c_1$ and $c_2$ are solved as:

$$\begin{cases} c_1(\phi) = \frac{\int_\Omega I(x) \cdot H(\phi)dx}{\int_\Omega H(\phi)dx} \\ c_2(\phi) = \frac{\int_\Omega I(x) \cdot (1 - H(\phi))dx}{\int_\Omega (1 - H(\phi))dx} \end{cases} \tag{10}$$

and

$$H_\varepsilon(z) = \frac{1}{2}\left(1 + \frac{2}{\pi}\arctan\left(\frac{z}{\varepsilon}\right)\right) \tag{11}$$

The total variation level set formulation is defined as:

$$\frac{\partial\phi}{\partial t} = spf(I(x)) \cdot \left(div\left(\frac{\nabla\phi}{|\nabla\phi|}\right) + \alpha\right)|\nabla\phi| + \nabla spf(I(x)) \cdot \nabla\phi \tag{12}$$

where $\alpha$ is the balloon force, which controls the closed contour shrinking or expanding.

## LRFI model

Liu et al. utilized local regional fitting information to propose an improved level set method, which can differentiate the noise and boundary points of the image to be segmented. In this model, two innovations are proposed: first, a controllable velocity coefficient was proposed to accelerate the curve convergence speed, and second, a new edge stop function was constructed to enhance the performance of the segmentation model. The velocity function was shown as:

$$v(x) = \alpha e^{-\beta |f_{in}(x) - f_{out}(x)|} + k \tag{13}$$

where $f_{in}(x)$ and $f(x)$ are local regional fitting means of image pixels inside and outside of the closed contour, respectively, and $\alpha$ and $k$ are two positive coefficients.

And, the edge stop function (ESF) was defined as:

$$g(x) = \frac{1}{1 + f / (\sigma_{in}^2(x) + \sigma_{out}^2(x) + 1)} \tag{14}$$

According to the principle of the variational method, the curve iteration function of the model is described as follows:

$$\frac{\partial \phi}{\partial t} = \mu \, div(d_p(|\nabla \phi|)\nabla \phi) + \lambda \delta(\phi) div\left(g(x)\frac{\nabla \phi}{|\nabla \phi|}\right) + \delta(\phi)v(x)g(x) \tag{15}$$

Compared with the DRLSE model, the function $v(x)$ of the LRFI model can better make the closed curve converge along the object boundaries, and the ESF improves the accuracy of the numerical calculation. The LRFI model demonstrates good performance in segmenting noisy images.

## Summary

In summary, the four segmentation models based on the level set method have made great contributions to image segmentation with intensity inhomogeneity. As a typical segmentation model based on the level set, the CV model has the merit of high convergence speed if segmenting two phase images with intensity homogeneity. The DRLSE model normalizes the movement of the curve during the curve evolution process and improves the accuracy of the numerical solution. However, this model has poor performance in segmenting weak boundaries and high noise images. In the LVC model, the SPF is used to control the direction of the curve evolution which further improves the speed of iteration. The model improves the efficiency of the segmentation algorithm. Although this model is more efficient than the DRLSE model, this model loses its advantages when the image to be segmented has high intensity inhomogeneity and is less efficient when segmenting severe high-noise images. The LRFI model improves the efficiency of segmenting noisy images; however, it does not perform well in segmenting fuzzy boundary images.

Actually, the above segmentation models have their own advantages and disadvantages. How to make full use of the local information of an image is the focus of improving the accuracy of the segmentation model. In the proposed model, we utilize the statistical information of the local region to construct a weighted pressure force function, which adaptively shrinks and expands the closed contour. In addition, an enhanced distance regularized level set method is proposed to improve the speed of the segmentation algorithm, and to avoid the process of reinitialization.

## The proposed model

Through the analysis of the above model, two core issues need to be solved: first, how to freely shrink and expand the closed contour by using the local information of an image; second, how to improve the accuracy of the segmentation model solution. Through in-depth research on the energy-based segmentation model in the previous work, this paper proposes a novel segmentation model based on local energy. Specifically, in this section, we present the details of the proposed weighted region-based ACM, which is based on the techniques of local statistical theory, curve evolution and the level set method. In the model, we exploit the statistical information inside and outside the closed curve contour of the image to construct a new weighted pressure force (WPF) function, which can effectively make the closed contour shrink outside of the object or expand inside of the object. Based on the WPF, we propose an enhanced distance regularized LSM (EDRLSM), which uses a polynomial function instead of a trigonometric function to further improve the speed of the algorithm. It improves the speed of segmentation by avoiding the process of reinitialization and the efficiency of the segmentation model.

### The WPF function

Inspired by the LVC model, we construct the WPF function as follows:

$$wpf = \frac{I(x) - (\lambda_1 c_1 + \lambda_2 c_2)/(\lambda_1 + \lambda_2)}{\max(|I(x) - (\lambda_1 c_1 + \lambda_2 c_2)/(\lambda_1 + \lambda_2)|)} \tag{16}$$

where $c_1$ and $c_2$ are defined in Eq (10), and $\lambda_1$ and $\lambda_2$ are two control parameters that control the inside or outside of the closed contour.

The significance of Eq (16) is that it can modulate the signs of the balloon forces inside and outside of the closed contour: the closed contour can shrink when it is outside the object and the closed contour can expand when it is inside the object. Compared with the LVC model, the advantage of the proposed model is that WPF improves the accuracy of the model's segmentation of intensity inhomogeneity images by restricting the unevenness of the intensity mean value inside and outside the closed curve, and improves the model's anti-noise ability.

Substituting the WPF function in Eq (16) for the *ESF* in Eq (7), the variational level set formulation of the proposed model is as follows:

$$\frac{\partial \phi}{\partial t} = \mu div(d_p(|\nabla \phi|)\nabla \phi) + \alpha \cdot g\delta_\varepsilon(\phi) + \lambda \delta_\varepsilon(\phi) div\left( wpf(I(x)) \cdot \frac{\nabla \phi}{|\nabla \phi|} \right) \tag{17}$$

where $div(.)$ is called the divergence operator. $D = \mu dp(|\nabla \phi|)$ is called the diffusion rate which is smooth and bounded. Therefore, it ensures that the curve adaptively increases or decreases to maintain the ideal shape of the implicit function $\phi$.

In the right side of Eq (17), the first item represents the regularized term, which regularizes the LSF to be a signed distance function; the second item is a weighted area of the closed region, which speeds up the motion of the zero-level contour; the third item is called the length penalty of the closed curve, which is able to be located at the object boundaries when the energy function is minimized.

Different from the DRLSE model, we use a more efficient diffusion rate to improve the efficiency of algorithm execution. The function constructed in our previous work [6] is as follows:

$$d_p(s) = \begin{cases} (s^2 - 1)^2 + 2s^2(s^2 - 1), & 0 \le s \le 1 \\ 1 - {}^1\!/_s, & s > 1 \end{cases} \tag{18}$$

In terms of the execution efficiency of the algorithm, compared with other models, the advantage of this function is that it has a smaller amount of calculation, which can improve the efficiency of curve iteration.

## Implementation of EDRLSM

Reinitialization is a complex and time-consuming process in traditional level set methods. Reinitialization is necessary to assure the accuracy of the numerical calculation. How and when to perform this operation is still a major problem. To solve the problem, the proposed model utilizes the regularized term to intrinsically maintain the stability of the evolution during the level set evolution. The weighted area term is introduced to accelerate the velocity of curve iteration. The length penalty term is based on the region statistical information, which is important for images with intensity inhomogeneity. Compared with the LVC model, its advantage lies in the ability to flexibly adjust the influence of the intensity mean inside and outside of the closed region on the segmentation according to the intensity inhomogeneity of the image.

The implementation of the proposed model is described as follows:

Step 1: Initialization. Initialize the control parameters $\lambda_1$, $\lambda_2$, $\mu$, $\alpha$ and $\lambda$. We initialize the LSF $\phi$ to a function $\phi_0$ shown as follows:

$$\phi(x, t = 0) = \begin{cases} -2, x \in \Omega_0 - \partial\Omega_0, \\ 0, \ x \in \partial\Omega_0 \\ 2, \ x \in \Omega - \Omega_0 \end{cases} \tag{19}$$

where $\Omega_0$ is the domain of the initialization function of the LSF and $\partial\Omega_0$ is the boundary of $\Omega_0$.

Step 2: Compute. Compute the intensity mean of the image. In the *WPF* function, the values of $c_1$ and $c_2$ defined in Eq (10) should be calculated to obtain the value *WPF*.

Step 3: Iteration. Start the loop iteration process according to Eq (17).

Step 4: Judgement. Determine whether the iterative process is over; if it is, return the result, if not, jump to Step 2.

## Advantage of the EDRLSM model over the other models

The advantage of the proposed model lies in the WPF function we constructed, which not only suppresses the influence of noise, thereby improving the accuracy of segmentation, but also increases the speed of the segmentation algorithm.

Compared with the classic CV model, the EDRLSM utilizes local information of the image to construct the WPF, and the proposed model establishes an energy function based on the local intensity information of the image, which greatly improves the accuracy of image segmentation with intensity inhomogeneity. Compared with the DRLSE model and the GAC model, the proposed model utilizes the image statistical information to construct an energy functional model, as a result, the proposed model is more robust to noise. Our model can successfully segment images with weak boundaries. The segmentation accuracy of our model is similar to that of the LVC model, but our model has good stability in terms of efficiency. For details, see the experimental section below.

## Experimental results

This section shows experiments to demonstrate the effectiveness of the proposed model for both synthetic and real images. The proposed model is compared with the state-of-the art

segmentation model based on level set methods to validate the effectiveness and robustness of our model. The proposed model is implemented in MATLAB R2018b on a 2.3 GHz Intel and 8.0GB RAM computer.

## Parameter setting

In the proposed model, main parameters used in the experiments are given in Table 1. Based on a large amount of experimental experience, we have obtained the initial values of the following parameters. What needs to be emphasized is that the values of $\lambda_1$ and $\lambda_2$ need to be selected according to the scale of intensity inhomogeneity of the image. In this experiment, the default values of these two parameters are both 1. In each experiment, the control parameters are set as follows:

## Qualitative evaluation

In the first experiment, we compare the proposed model with the LVC, GAC and DRLSE models in segmenting a synthetic image with noise. As shown in Fig 1, the image has the characteristics of fuzzy boundaries and multiple sharp corners, which brings greater difficulty to segmentation. The GAC, as a typical representative of energy-based models, has made a certain contribution to image segmentation. However, when the edges of the image are blurry or sharp corners, the segmentation accuracy of GAC model is significantly reduced. The regularization idea in The DRLSE model better standardizes the iteration of the curve and improves the accuracy of the solution, but the segmentation accuracy of the fuzzy boundary image is not high. As shown in Fig 1, compared with the GAC and DRLSE models, the segmentation accuracy of this model is higher at the fuzzy angles of the image. As a representative edge-based LSM, the GAC and DRLSE models perform poorly, mainly because the synthetic image is affected by noise and blurred boundaries.

To further verify the advantage of the proposed model, we compare it with three typical active contour models, including the CV model, DRLSE model, LSACM model and LVC model on three medical images with intensity inhomogeneity. In the proposed model, the controllable parameters are set to $\lambda_1 = 3$, $\lambda_2 = 2$, $\mu = 0.2$, $\alpha = 1.5$, $\lambda = 5$ and $\varepsilon = 1.5$. For a fair comparison, we use the same initialization information and set the same controllable parameters to the default values. The segmentation results are shown in Fig 2.

As shown in Fig 2, the segmentation results show that there is a certain gap between the five models. Specifically, the CV model is better for segmenting medical images with a small intensity level, but it has a poor segmentation effect for images with fuzzy boundaries. The main reason is that the construction of the model is based on the global information model, which is not suitable for the image segmentation with weak boundary images. The DRLSE model ensures the stability of LSF under the premise of ensuring the accuracy of segmentation. However, as shown in Fig 2, when the image's intensity inhomogeneity is high, the segmentation effect of this model is not good. For the brain image, the LSACM has a better segmentation result than that of the CV model and DRLSE model. However, the model is highly sensitive to initial information.

**Table 1. Value information table of each parameter.**

| Names | Values | Parameter Description |
|---|---|---|
| $\lambda_1$ | 1.0 | the control parameter inside the closed contour |
| $\lambda_2$ | 1.0 | the control parameter outside the closed contour |
| $\mu$ | 0.2 | coefficient of the diffusion rate |
| $\alpha$ | -3 | coefficient of the weighted area term |
| $\lambda$ | 5 | coefficient of the weighted length term |

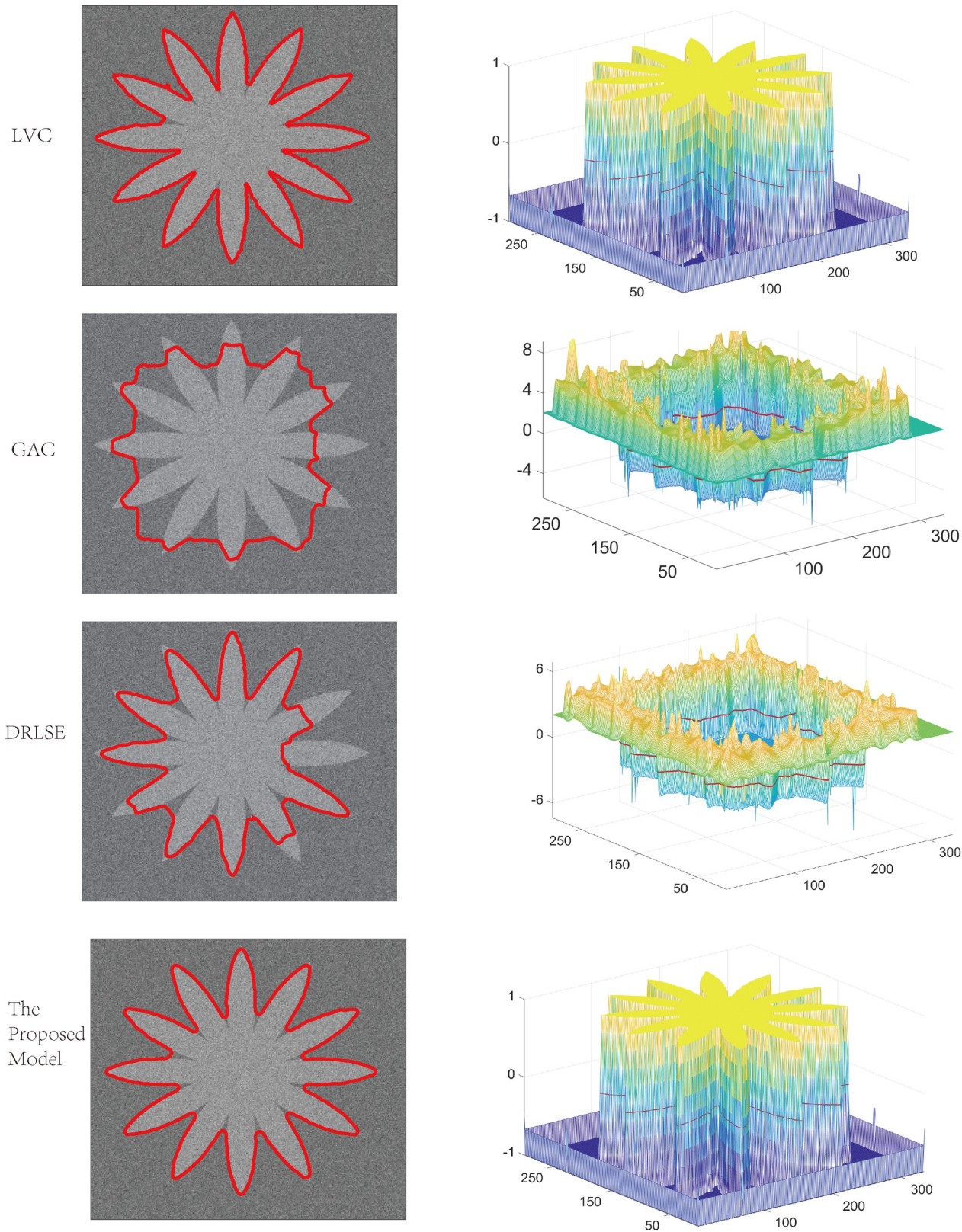

**Fig 1. Comparisons of the proposed model with the LVC, GAC, DRLSE models.**

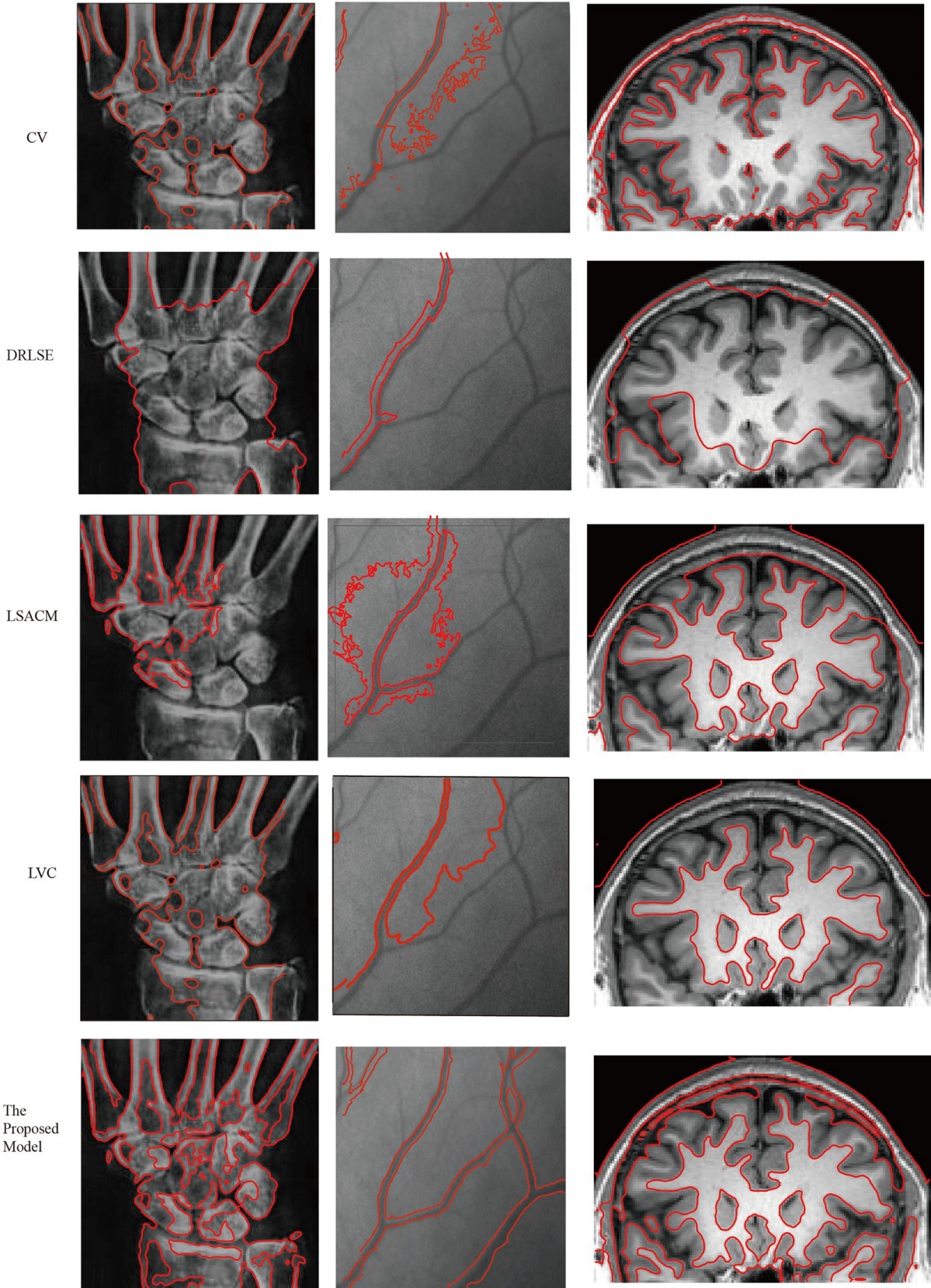

**Fig 2. Segmentation results of the proposed model and CV, DRLSE, LSACM, LVC models.**

Compared with the other models, the LVC model is the best in terms of segmentation speed, and the segmentation results are better. However, the generalization ability of this model is weak. For example, the segmentation effect of the model for the second image is poorer than that of the proposed model. Although the proposed model is not optimal in terms of segmentation speed, in terms of segmentation results, the segmentation effect of the model is best as shown in Fig 2. In the proposed model, the statistical information inside and outside the closed curve contour is utilized to construct the weighted pressure force, which greatly improves the accuracy of image segmentation with intensity inhomogeneity. Therefore, by comparing multiple models, the proposed model performs better in the accuracy of segmenting medical images.

Brain tissue segmentation has always been a research hotspot and difficulty in medical image segmentation. As shown in Fig 3, this experiment is mainly used to compare the accuracy of segmented brain tissue images. The proposed model makes full use of the local information of the image to construct the model and obtains a good segmentation effect. Specifically, Experiment 3 applied segmentation models to real medical images from slices of cerebral tissue taken via MRI. Fig 3 depicts the performance comparison of the proposed model on the brain images with the standard models including the LIF model, the LSACM model and the LVC model. The segmentation of brain MRI slices of cerebral tissue is challenging due to poor spatial resolution, low tissue contrast and severe intensity variability. As shown in Fig 3, column (a) shows the segmentation results of the LIF. This model utilized the image local statistics information and successfully segmented images with intensity inhomogeneities. Fig 3 shows that the results were good, except for the poor segmentation accuracy at the blurred boundaries of the last two image from top to bottom. The LSACM, proposed by Zhang et al., constructed a mapping method by using Gaussian distributions of different means and variances and mapped the original image into another domain, which was easy to segment.

As shown in Fig 3, the segmentation result is better than that of the LIF model. However, the hypothesis is limited, and the segmentation result is poor as shown in the third image. The LVC cannot segment the detailed part of the image. Therefore, through comparison of the segmentation results, it can be inferred that the segmentation accuracy of this model is better than that of the other three models.

In order to further demonstrate the efficiency of the proposed model, we test the proposed model for two images with severe intensity inhomogeneity shown as in Fig 4. The heart image has the characteristics of high noise and low contrast that brings certain challenges to image segmentation. As shown in Fig 4, the CV model cannot correctly obtain the boundary of the image to be segmented and it results in poor segmentation results. Compared with the LCV model, this model has a higher segmentation accuracy for multiple targets, and can quickly segment two parts of the heart based on local information. The segmentation effect of the LVC and LRFI models is not as good as that of the proposed model.

## Quantitative analysis

In the subsection, several typical synthetic image datasets and Berkeley segmentation datasets are tested to further demonstrate the capability of the proposed model. In all quantitative analysis methods, the Jaccard similarity coefficient is an evaluable method that can measure the similarity between finite sample sets [28–30]. Specifically, we assume that $S_1$ and $S_2$ represent the segmented object region and the ground truth, respectively. The definition is as follows:

$$JS(S_1, S_2) = \frac{|S_1 \cap S_2|}{|S_1 \cup S_2|} \tag{20}$$

LIF                        LSACM                        LVC                    The Proposed Model

**Fig 3. Comparisons of the proposed model with the LIF, the LSACM and the LVC on brain images.**

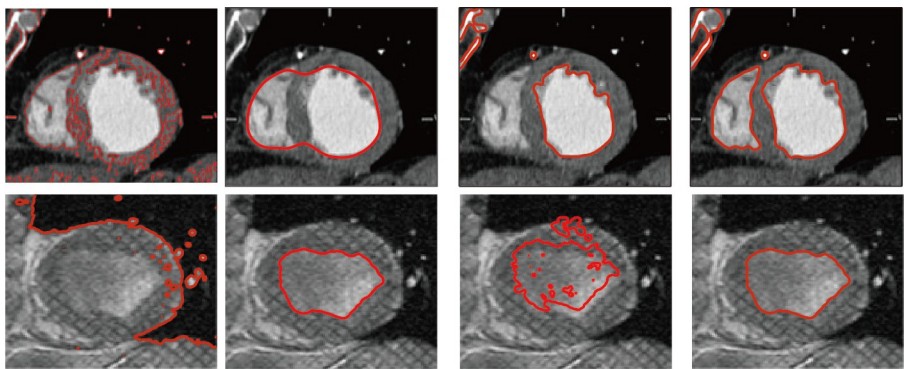

**Fig 4. Comparisons of the proposed model with the CV, the LVC, the LRFI on heart images.**

**Fig 5. Comparisons of the proposed model with the CV model and the LVC model.**

Next, we use it to compare the proposed model with state-of-the-art models, including the CV, LVC and LSACM models.

Fig 5 shows the comparison of the segmentation results of an image and its Gaussian noise image with a variance of 0.04 obtained by the three models, including the CV model, LVC model and proposed model. As a classic segmentation algorithm, the CV model performs well in image segmentation, however, when segmenting high-noise images, the segmentation effect of this model is slightly inferior. The main reason is that this model is based on global information, which leads to its poor anti-noise ability. The LVC model performs better in noisy image segmentation. It has a certain anti-noise ability and is better than the CV model in terms of segmentation speed and accuracy. The proposed model has a stronger anti-noise ability than the CV model and the LVC model.

Table 2 shows the JS comparison results of the proposed model with the CV model and the LVC model. For the images in Table 2, we explain that image 1 is the original image and image 2 is its noisy image. As shown in Fig 5 and Table 2, the CV model, the LVC model and the proposed model show little difference when segmenting the original image, however, when these models segment the noisy image, the proposed model performs better than the other models. The main reason is that the proposed model uses WPF to flexibly constrain the influence of the intensity mean value inside and outside the closed region, which can effectively segment intensity inhomogeneous images.

Fig 6 shows the comparison results of this model with the CV, LVC and LSACM models. These images come from the Berkeley segmentation dataset. For clarity, we name the images in Fig 6 image 3, image 4, image 5 and image 6 from top to bottom. These images are all significantly representative, in which the background of image 3 shows a certain degree of intensity inhomogeneity; image 4 is a type of typical texture image; image 5 does not have edge details except for uneven background illumination; and image 6 is a noisy image of image 5 with an added variance of 0.04, which makes segmentation more difficult. As shown in Fig 6, the first column represents the ground truth provided by the Berkeley dataset, and the second to fifth columns represent the results of the segmentation by CV, LVC, LSACM, and the model in this article, respectively. Table 2 shows the comparisons of JS coefficients of Fig 6, including CV, LVC, LSACM and the proposed method. To compare multiple dimensions, we compare the iteration times of these models as shown in Fig 7. Obviously, the proposed model has a certain

**Table 2. Comparisons of Jaccard similarity coefficient of Fig 5.**

| Models | JS of Image 1 | JS of Image 2 |
|---|---|---|
| CV | 0.99001 | 0.76323 |
| LVC | 0.99102 | 0.97963 |
| The Proposed Model | 0.99116 | 0.99002 |

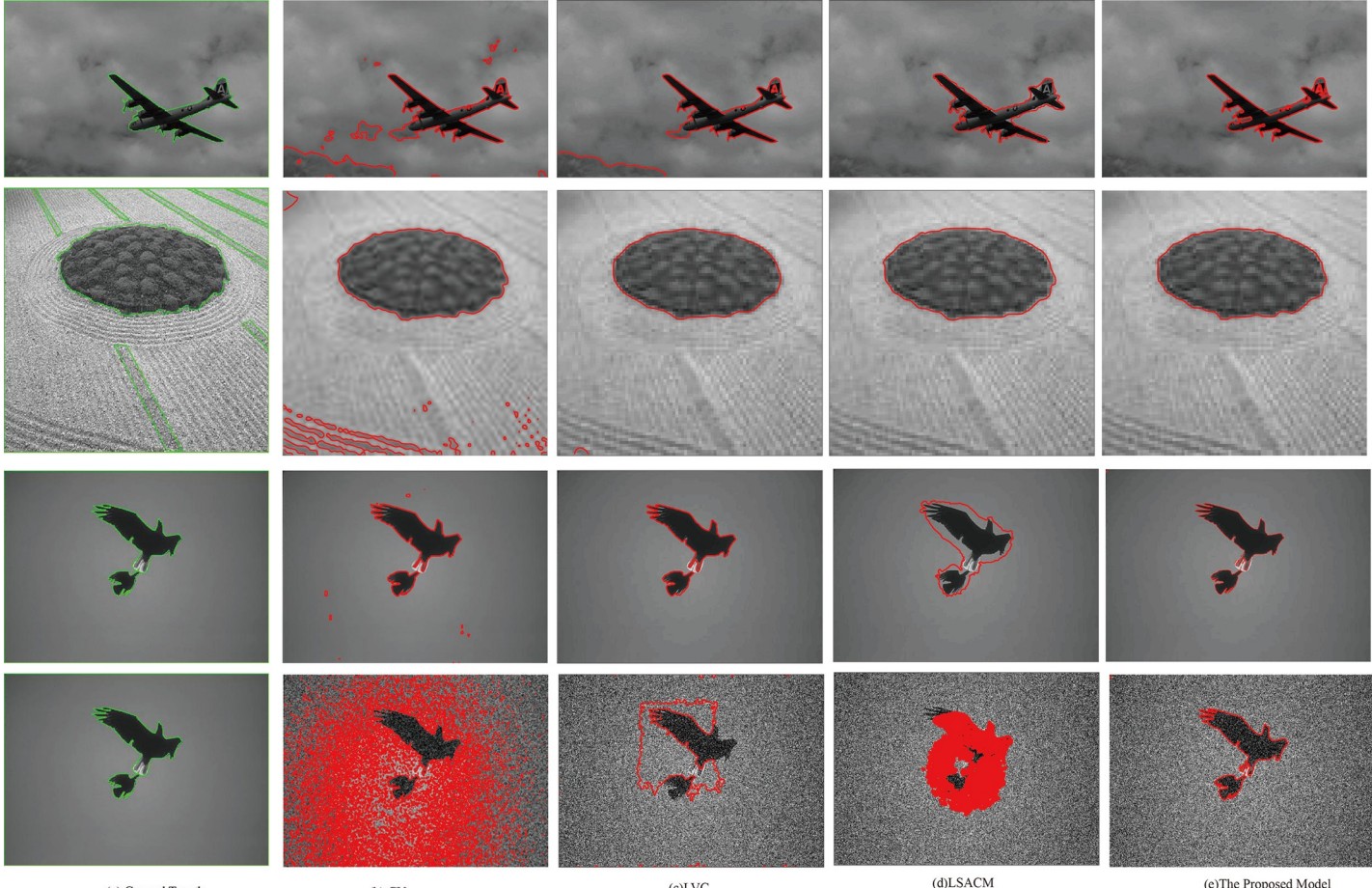

**Fig 6.** Comparisons of the proposed model with the CV model the LVC and the LSACM, they should be listed as: (a) The ground truth; (b) Segmentation results of the CV model; (c) Segmentation results of the LVC model; (d) Segmentation results of the LSACM;(e) Segmentation results of the proposed model.

anti-noise ability compared with the other three models. The execution speed of the LVC model is the fastest compared to the other segmentation models. Although the segmentation effect of the CV model is not good, this model has the best stability. Because the sizes of images 3, 5 and 6 are relatively large, LSACM takes longer to segment large-size images than the other three models as shown in Fig 7. Therefore, as shown in Fig 6, Table 3 and Fig 7, the LVC model and the proposed model are superior to the other two models in terms of segmentation accuracy and segmentation efficiency.

## Conclusions and future work

In this paper, we proposed a weighted region-based active contour model. The model exploits the statistical information inside and outside the closed curve contour of the image to construct a new weighted pressure force (WPF) function. The WPF function was utilized to modulate the signs of the balloon forces inside and outside of the closed contour, aiming for contraction or expansion freely of the closed contour. In addition, a regularization term was adopted to ensure the stability of the curve evolution and reduce reinitialization in curve evolution. Extensive experiments on medical and natural images demonstrate that the proposed model is suitable for segmenting images with intensity inhomogeneity with high accuracy as

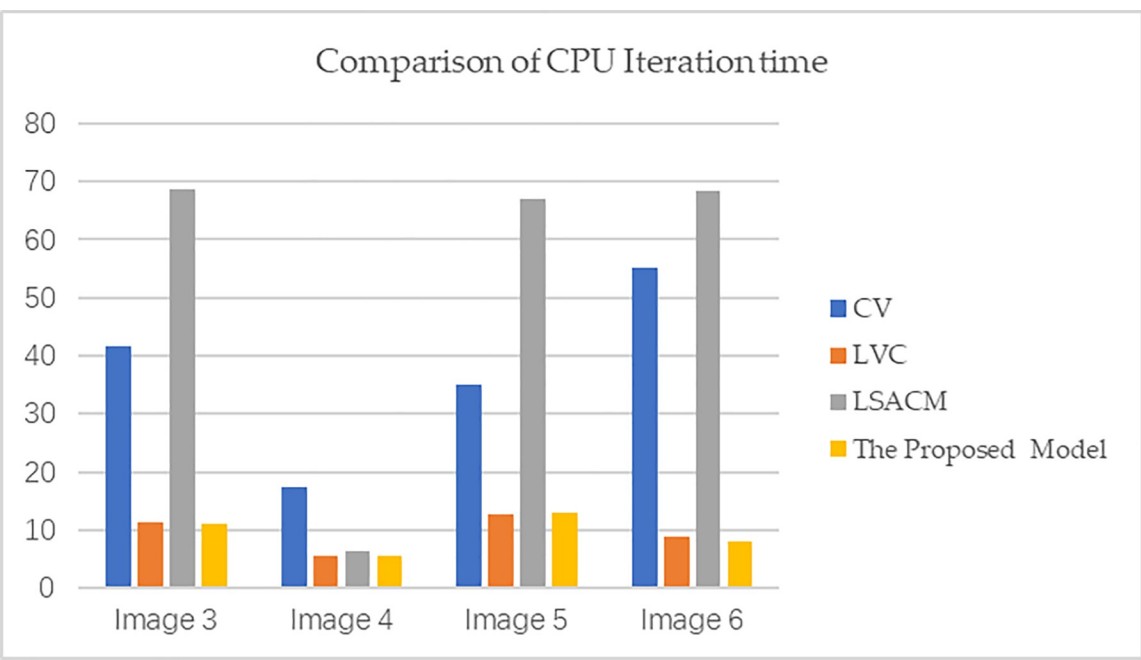

**Fig 7. Comparison of CPU iteration time of the proposed model with CV, LVC and LSACM.**

**Table 3. Comparisons of Jaccard similarity coefficient of Fig 6.**

| Models | JS of Image 3 | Js of Image 4 | Js of Image 5 | JS of Image 6 |
|--------|---------------|---------------|---------------|---------------|
| CV | 0.93711 | 0.92762 | 0.98001 | 0.32981 |
| LVC | 0.97832 | 0.93561 | 0.98122 | 0.76553 |
| LSACM | 0.99521 | 0.93572 | 0.90211 | 0.60001 |
| The Proposed Model | 0.99601 | 0.93610 | 0.98712 | 0.97602 |

compared to the other classic models. Moreover, the model seems to be more robust to severe noise levels and has less dependence on initialization information.

In terms of future work, we will continue our research at least three areas.

(1) We will try to improve the performance of the segmentation model using intelligent optimization methods. (2) We will expand the application fields of the proposed model, including semantic segmentation, edge detection and other related fields [31–39]. (3) We will study algorithm evaluation methods to objectively evaluate various algorithms in multiple dimensions.

## Supporting information

**S1 File. This is the description of the data set.**
(PDF)

## Acknowledgments

We would like to thank all the anonymous reviewers for their valuable comments.

## Author Contributions

**Funding acquisition:** Haiping Yu.

**Methodology:** Haiping Yu.

**Software:** Ping Sun.

**Supervision:** Fazhi He.

**Writing – review & editing:** Zhihua Hu.

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
