## [Decision Letter · Decision Letter 0]

11 May 2021

PONE-D-21-13529

A weighted region-based level set method for image segmentation with intensity inhomogeneity

PLOS ONE

Dear Dr. Yu,

Thank you for submitting your manuscript to PLOS ONE. After careful consideration, we feel that it has merit but does not fully meet PLOS ONE’s publication criteria as it currently stands. Therefore, we invite you to submit a revised version of the manuscript that addresses the points raised during the review process.

We look forward to receiving your revised manuscript.

Kind regards,

Zhifan Gao

Academic Editor

PLOS ONE

Journal Requirements:

2. Please include the data sources used in the Data availability statement and Methods section. We note that the origin of the images does not seem to be referenced.

"We would like to thank all the anonymous reviewers for their valuable comments. This work is

supported by the Natural Key Research and Development Project of China (Grant No. 2016YFC0106305) and Hubei

Provincial Natural Science Foundation of China (Grant No. 2019CFB797)."

"Hubei Provincial Natural Science Foundation of China (Grant No. 2019CFB797). "

4. Please amend the manuscript submission data (via Edit Submission) to include author Ping Sun.

Reviewers' comments:

Reviewer's Responses to Questions

**Comments to the Author**

1. Is the manuscript technically sound, and do the data support the conclusions?

Reviewer #1: Partly

Reviewer #2: Partly

2. Has the statistical analysis been performed appropriately and rigorously? 

Reviewer #1: No

Reviewer #2: Yes

3. Have the authors made all data underlying the findings in their manuscript fully available?

Reviewer #1: No

Reviewer #2: Yes

4. Is the manuscript presented in an intelligible fashion and written in standard English?

Reviewer #1: No

Reviewer #2: Yes

5. Review Comments to the Author

Reviewer #1: In this paper, an image segmentation method based on weighted region-based level set method is proposed. It is based on the techniques of local statistical theory, level set theory and curve evolution.

1- The proposed method is not clear enough, and needs more explanation.

2- Previous work section needs more recent related work, and research gap should be explained.

3- Author must compare proposed method with more image segmentation methods such as sematic segmentation.

4- Experimental results should have more explanation.

5- The presentation and writing of this work need major revision.

6- Conclusion part needs more revision, and should be rewritten.

7- References need major revision, for example references 2, 5, 30 , 31, 35.

8- The paper should be in the Plos one format not IEEE Access.

9- I think the authors should spend some time in revision justifying why this work is worthy of publication.

Reviewer #2: This manuscript proposes a level-set-based method for image segmentation. The manuscript is clear, straightforward, easy to follow. However, I would like to draw the author's attention to the following major concerns:

1)Overall, the motivation is not introduced well, where the challenges should be described before the contributions. I recommend the authors to employ certain intuitive examples to elaborate the novelties of the proposed work.

2)The paper does not explain clearly its advantages with respect to recent deep-learning literature: it is not clear what is the novelty and contributions of the proposed work: does it propose a new method? Or does the novelty only consists in the application?

3) What is the motivation of the proposed work? Research gaps, objectives of the proposed work should be clearly justified. The authors should consider more recent research done in the field of their study

4) Authors should add more details about the implementation of the code to perform the analysis and the dateset involved in this task.

5) Considering that deep-learning-based methods (Such as Mask RCNN, Unet) have shown impressive performance in image segmentation, the advantages and disadvantages of the proposed method should be discussed more clearly, and presented in the experiment.

6) The authors should do a more thorough literature survey. Just to name a few:

- Ronneberger, O., Fischer, P., & Brox, T. (2015, October). U-net: Convolutional networks for biomedical image segmentation. In International Conference on Medical image computing and computer-assisted intervention (pp. 234-241). Springer, Cham.

- Xu, C., Xu, L., Gao, Z., Zhao, S., Zhang, H., Zhang, Y., ... & Li, S. (2018). Direct delineation of myocardial infarction without contrast agents using a joint motion feature learning architecture. Medical image analysis, 50, 82-94.

- Minaee, S., Boykov, Y. Y., Porikli, F., Plaza, A. J., Kehtarnavaz, N., & Terzopoulos, D. (2021). Image segmentation using deep learning: A survey. IEEE Transactions on Pattern Analysis and Machine Intelligence.

- Sultana, F., Sufian, A., & Dutta, P. (2020). Evolution of image segmentation using deep convolutional neural network: a survey. Knowledge-Based Systems, 201, 106062.

7) There are some grammar errors and typos. I suggest the authors make an solid, overall proofreading.

6. PLOS authors have the option to publish the peer review history of their article (what does this mean?). If published, this will include your full peer review and any attached files.

Reviewer #1: No

Reviewer #2: No

---

## [Author Response · Author response to Decision Letter 0]

15 Jul 2021

A list of responses to the comments

Manuscript Number: PONE-D-21-13529

Manuscript Title:“A weighted region-based level set method for image segmentation with intensity inhomogeneity”

Dear Reviewers and Editor,

We totally agree with the valuable comments, which is a great help not only to improve the quality of this manuscript but also to instruct our research in future! We also thank editor for giving us an opportunity to revise the manuscript!

According to reviewers’ comments, we have made a careful revision. In this response letter, we will try our best to list the details how each section is revised so as to illustrate the revision clearly.

Sincerely yours,

Haiping Yu, on behalf of the authors

Email: seapingyu@outlook.com

Legend

Black color: reviewers’ comments texts

Black color: original texts in original version

Blue color: our explanation texts to comments

Red color: the modified (or added) texts in revised version

 

COMMENTS FOR THE AUTHOR:

Reviewer #1: In this paper, an image segmentation method based on weighted region-based level set method is proposed. It is based on the techniques of local statistical theory, level set theory and curve evolution.

Thanks very much for kind comment! 

Accordingly, based on your following detail comments, we improve the manuscript as much as possible in revised version one by one as follows.

1- The proposed method is not clear enough, and needs more explanation.

Answer:

Thank you for your valuable comments! Your comments are very instructive and helpful for our future work. 

First, the author apologize that the proposed method is not clear enough! 

 Second, in revised version, we have carefully improved the quality of the proposed method.

The detail changes in revised version are as follows:

In the last paragraph of Introduction：

In this paper, we propose a weighted region-based level set method to segment images with intensity inhomogeneity. In the model, first, a new weighted pressure force (WPF) is proposed to adaptively modulate the contractility of the balloon forces inside and outside of the closed contour by restricting the unevenness of the intensity mean value inside and outside the closed curve; second, a faster and smoother regularization term is utilized to ensure the stability of the curve evolution during the process of solving; finally, the WPF is integrated into the level set framework to improve the accuracy of segmentation images with intensity inhomogeneity.

In Previous works, we added a new subsection to elaborate the idea of the proposed segmentation model by comparing with other standard models.

Summary

In summary, the four segmentation models based on the level set method have made great contributions to image segmentation with intensity inhomogeneity. As a typical segmentation model based on the level set, the CV model has the merit of high convergence speed if segmenting two phase images with intensity homogeneity. The DRLSE model normalizes the movement of the curve during the curve evolution process and improves the accuracy of the numerical solution. However, this model has poor performance in segmenting weak boundaries and high noise images. In the LVC model, the SPF is used to control the direction of the curve evolution which further improves the speed of iteration. The model improves the efficiency of the segmentation algorithm. Although this model is more efficient than the DRLSE model, this model loses its advantages when the image to be segmented has high intensity inhomogeneity and is less efficient when segmenting severe high-noise images. The LRFI model improves the efficiency of segmenting noisy images; however, it does not perform well in segmenting fuzzy boundary images.

Actually, the above segmentation models have their own advantages and disadvantages. How to make full use of the local information of an image is the focus of improving the accuracy of the segmentation model. In the proposed model, we utilize the statistical information of the local region to construct a weighted pressure force function, which adaptively shrinks and expands the closed contour. In addition, an enhanced distance regularized level set method is proposed to improve the speed of the segmentation algorithm, and to avoid the process of reinitialization.

2- Previous work section needs more recent related work, and research gap should be explained.

Answer:

Thank you very much for your valuable comments, which is a great help not only to improve the quality of this manuscript but also to instruct our research in future!

First, we added two sections to the previous works section to introduce two recent classic segmentation models. 

Second, we added a summary to the previous works section to summarize the previous work and explain the research gap.

The detail changes in revised version are as follows:

In the “Previous works” section, two subsections are added to the first subsection (CV model) and the fourth subsection (LRFI model), respectively.

Previous works

CV model

The CV model, as a classic typical energy-based segmentation model, is proposed to segment two-phase images, based on the assumption that the target is segmented and the background is intensity inhomogeneous. Let an image I(x) on the image domain �, the energy function is defined as:

 ……

 The CV model can well segment images with intensity inhomogeneity, however, it has more sensitivity to initialization information and low efficiency while segmenting images with high noise and severe intensity inhomogeneity.

……

LRFI model

Liu et al. utilized local regional fitting information to propose an improved level set method, which can differentiate the noise and boundary points of the image to be segmented. In this model, two innovations are proposed: first, a controllable velocity coefficient was proposed to accelerate the curve convergence speed, and second, a new edge stop function was constructed to enhance the performance of the segmentation model. The velocity function was shown as:

 ……

where fin(x) and fout(x) are local regional fitting means of image pixels inside and outside of the closed contour, respectively, and � and k are two positive coefficients.

And, the edge stop function (ESF) was defined as:

 ……

According to the principle of the variational method, the curve iteration function of the model is described as follows:

 ……

Compared with the DRLSE model, the function v(x) of the LRFI model can better make the closed curve converge along the object boundaries, and the ESF improves the accuracy of the numerical calculation. The LRFI model demonstrates good performance in segmenting noisy images.

In the last subsection, summary is added to summarize the previous work and explain the research gap.

Summary

In summary, the four segmentation models based on level set method have made great contributions to images segmentation with intensity inhomogeneity. As a typical segmentation model based on level set, The CV model has the merit of high convergence speed if segmenting two phase images with intensity homogeneity. The DRLSE model normalizes the movement of the curve during the curve evolution process and improves the accuracy of the numerical solution. However, this model has poor performance in segmenting weak boundaries and high noise images. In the LVC model, the SPF is used to control the direction of the curve evolution which further improves the speed of iteration. The model improves the efficiency of the segmentation algorithm. Although this model is more efficient than the DRLSE model, this model loses its advantages when the image to be segmented has high intensity inhomogeneity and is less efficient when segmenting severe high-noise images. The LRFI model improves the efficiency of segmenting noisy images, however, it does not perform well in segmenting fuzzy boundary images.

Actually, the above segmentation models have their own advantages and disadvantages. How to make full use of the local information of the image is the focus of improving the accuracy of the segmentation model. In the proposed model, we utilize the statistical information of the local region to construct a weighted pressure force function, which adaptively shrink and expand the closed contour. In addition, an enhanced distance regularized level set method is proposed to improve the speed of segmentation algorithm, and to avoid the process of reinitialization.

3- Author must compare proposed method with more image segmentation methods such as sematic segmentation.

Answer:

Thank you very much for your valuable comments, which is a great help not only to improve the quality of this manuscript but also to instruct our research in future!

First, we have added two classic segmentation models to related works. These models are the same type of segmentation method as the model proposed in this manuscript, so there are comparable to the proposed model.

Second, in addition to the above method, we also added comparative experiments with other classic methods in the experimental section.

The detail changes in revised version are as follows:

In the section of “Previous works”, two classic segmentation models are added to the related works.

Previous works

CV model

 The CV model, as a classic typical energy-based segmentation model, is proposed to segment two-phase images, based on the assumption that the target is segmented and the background is intensity inhomogeneous. Let an image I(x) on the image domain �, the energy function is defined as:

 ……

 According to the variational method, the evolution equation of the curve is expressed as follows:

 …… 

 The CV model can segment images with intensity inhomogeneity; however, it has more sensitivity to initialization information and low efficiency while segmenting images with high noise and severe intensity inhomogeneity.

……

LRFI model

Liu et al. utilized local regional fitting information to propose an improved level set method, which can differentiate the noise and boundary points of the image to be segmented. In this model, two innovations are proposed: first, a controllable velocity coefficient was proposed to accelerate the curve convergence speed, and second, a new edge stop function was constructed to enhance the performance of the segmentation model. The velocity function was shown as:

 ……

According to the principle of the variational method, the curve iteration function of the model is described as follows:

 …… 

Compared with the DRLSE model, the function v(x) of the LRFI model can better make the closed curve converge along the object boundaries, and the ESF improves the accuracy of the numerical calculation. The LRFI model demonstrates good performance in segmenting noisy images.

we also added comparative experiments with other classic methods in the experimental section.

Figure 4. Comparisons of the proposed model with the CV, the LVC, the LRFI on heart images

 In order to further demonstrate the efficiency of the proposed model, we test the proposed model for two images with severe intensity inhomogeneity shown as in Figure 4. The heart image has the characteristics of high noise and low contrast that brings certain challenges to image segmentation. As shown in Figure 4, the CV model cannot correctly obtain the boundary of the image to be segmented and it results in poor segmentation results. Compared with the LCV model, this model has a higher segmentation accuracy for multiple targets, and can quickly segment two parts of the heart based on local information. The segmentation effect of the LVC and LRFI models is not as good as that of the proposed model.

4- Experimental results should have more explanation.

Answer:

Thank you very much for your valuable comments, which is a great help not only to improve the quality of this manuscript but also to instruct our research in future! 

It's our fault not to describe it clearly in the experiment part. We have done a lot of improvement work in revised version.

The detail changes in revised version are as follows:

In the section of “Experimental results”, we have analyzed the experimental results in detail to verify the advantages of this model over other models.

Experimental results

This section shows experiments to demonstrate the effectiveness of the proposed model for both synthetic and real images. The proposed model is compared with the state-of-the art segmentation model based on level set methods to validate the effectiveness and robustness of our model. The proposed model is implemented in MATLAB R2018b on a 2.3 GHz Intel and 8.0GB RAM computer. 

……

Qualitative evaluation

In the first experiment, we compare the proposed model with the LVC, GAC and DRLSE models in segmenting a synthetic image with noise. As shown in Figure 1, the image has the characteristics of fuzzy boundaries and multiple sharp corners, which brings greater difficulty to segmentation. The GAC, as a typical representative of energy-based models, has made a certain contribution to image segmentation. However, when the edges of the image are blurry or sharp corners, the segmentation accuracy of GAC model is significantly reduced. The regularization idea in The DRLSE model better standardizes the iteration of the curve and improves the accuracy of the solution, but the segmentation accuracy of the fuzzy boundary image is not high. As shown in Figure 1, compared with the GAC and DRLSE models, the segmentation accuracy of this model is higher at the fuzzy angles of the image. As a representative edge-based LSM, the GAC and DRLSE models perform poorly, mainly because the synthetic image is affected by noise and blurred boundaries.

……

Compared with the other models, the LVC model is the best in terms of segmentation speed, and the segmentation results are better. However, the generalization ability of this model is weak. For example, the segmentation effect of the model for the second image is poorer than that of the proposed model. Although the proposed model is not optimal in terms of segmentation speed, in terms of segmentation results, the segmentation effect of the model is best as shown in Figure 2. In the proposed model, the statistical information inside and outside the closed curve contour is utilized to construct the weighted pressure force, which greatly improves the accuracy of image segmentation with intensity inhomogeneity. Therefore, by comparing multiple models, the proposed model performs better in the accuracy of segmenting medical images.

Brain tissue segmentation has always been a research hotspot and difficulty in medical image segmentation. As shown in Figure 3, this experiment is mainly used to compare the accuracy of segmented brain tissue images. The proposed model makes full use of the local information of the image to construct the model and obtains a good segmentation effect. Specifically, Experiment 3 applied segmentation models to real medical images from slices of cerebral tissue taken via MRI. Figure 3 depicts the performance comparison of the proposed model on the brain images with the standard models including the LIF model, the LSACM model and the LVC model.

5- The presentation and writing of this work need major revision.

Answer:

Thank you very much for your valuable comments, which is a great help not only to improve the quality of this manuscript but also to instruct our research in future! 

After we carefully revised the manuscript based on expert reviews, then the manuscript was edited for proper English language, grammar, punctuation, spelling, and overall style by one or more of the highly qualified native English-speaking editors at AJE.

6- Conclusion part needs more revision, and should be rewritten.

Answer:

Thank you very much for your valuable comments, which is a great help not only to improve the quality of this manuscript but also to instruct our research in future! 

The detail changes in revised version are as follows:

Conclusions and future work

In this paper, we proposed a weighted region-based active contour model. The model exploits the statistical information inside and outside the closed curve contour of the image to construct a new weighted pressure force (WPF) function. The WPF function was utilized to modulate the signs of the balloon forces inside and outside of the closed contour, aiming for contraction or expansion freely of the closed contour. In addition, a regularization term was adopted to ensure the stability of the curve evolution and reduce reinitialization in curve evolution. Extensive experiments on medical and natural images demonstrate that the proposed model is suitable for segmenting images with intensity inhomogeneity with high accuracy as compared to the other classic models. Moreover, the model seems to be more robust to severe noise levels and has less dependence on initialization information. 

In terms of future work, we will continue our research at least three areas. 

(1) We will try to improve the performance of the segmentation model using intelligent optimization methods. (2) We will expand the application fields of the proposed model, including semantic segmentation, edge detection and other related fields [31-39]. (3) We will study algorithm evaluation methods to objectively evaluate various algorithms in multiple dimensions.

7- References need major revision, for example references 2, 5, 30, 31, 35.

Answer:

Thank you very much for your valuable comments, which is a great help not only to improve the quality of this manuscript but also to instruct our research in future!

First, the authors apologize that the reference format is incorrect.

Second, in revised version, we have carefully improved the references in accordance with the format requirements. We revised the references 2,5,30,31,35. We also have made major revision for all the references (old and added) in accordance with the format requirements.

The detail changes in revised version are as follows:

[1] Xie F, Bovik A C. Automatic segmentation of dermoscopy images using self-generating neural networks seeded by genetic algorithm. Pattern Recognition. 2013; 46(3): 1012-1019. doi: 10.1016/j.patcog.2012.08.012 

[2] Ren D , Jia Z , Yang J , et al. A Practical GrabCut Color Image Segmentation Based on Bayes Classification and Simple Linear Iterative Clustering. IEEE Access. 2017;18480-18487. doi: 10.1109/ACCESS.2017.2752221 

[3] Wang L, Li S, Chen R, et al. An Automatic Segmentation and Classification Framework Based on PCNN Model for Single Tooth in Micro CT Images. Plos One. 2016;11(6): e0157694. doi: 10.1371/journal.pone.0157694 PMID:27322421

[4] Ahmed I, Rehman S U, Khan I U, et al. A Hybrid Approach for Automatic Aorta Segmentation in Abdominal 3D CT Scan Images. Journal of Medical Imaging and Health Informatics.2020; doi:10.1166/jmihi.2021.3364

[5] Hu J, Wang H, Wang J, et al. SA-Net: A scale-attention network for medical image segmentation. PLoS ONE. 2021; 16(4): e0247388. doi:10.1371/journal.pone.0247388

[6] Karn, P. K., Biswal, B., Samantaray, S. R. Robust retinal blood vessel segmentation using hybrid active contour model. IET Image Processing. 2019; 13(3):440-450. doi:10.1049/iet-ipr.2018.5413

[7] Cai L, Long T, Dai Y, et al. Mask R-CNN-Based Detection and Segmentation for Pulmonary Nodule 3D Visualization Diagnosis. IEEE Access. 2020; 8:44400-44409. doi: 10.1109/ACCESS.2020.2976432

[8] Azizi, A., Elkourd, K., Azizi, Z. Robust active contour model guided by local binary pattern stopping function. Cybernetics & Information Technologies. 2017; 17(4):165-182. doi:10.1515/cait-2017-0047

[9] Yu, H., He, F., & Pan, Y. A novel segmentation model for medical images with intensity inhomogeneity based on adaptive perturbation. Multimedia Tools and Applications. 2019; 78(9): 11779-11798. doi:10.1007/s11042-018-6735-5

[10] Liang, Y., He, F., Zeng, X. 3d mesh simplification with feature preservation based on whale optimization algorithm and differential evolution. Integrated Computer Aided Engineering. 2020; 27(4):417-435. doi:10.3233/ICA-200641

[11] Chen, Y., He, F., Li, H., Zhang, D., Wu, Y. A full migration BBO algorithm with enhanced population quality bounds for multimodal biomedical image registration. Applied Soft Computing, Applied Soft Computing. 2020; 93. doi: 10.1016/j.asoc.2020.106335

[12] Osher, S., Sethian, J. A. Fronts propagating with curvature-dependent speed: Algorithms based on Hamilton-Jacobi formulations. Journal of Computational Physics. 1988; 79(1): 12-49. doi: 10.1016/0021-9991(88)90002-2

[13] Li, H., He, F., Chen, Y., Pan, Y. MLFS-CCDE: multi-objective large-scale feature selection by cooperative coevolutionary differential evolution. Memetic Computing. 2021; 13: 1-18. doi: 10.1007/s12293-021-00328-7 

[14] Yang, L., Yan, Q., Fu, Y., & Xiao, C. Surface reconstruction via fusing sparse-sequence of depth images. IEEE transactions on visualization and computer graphics. 2017; 24(2):1190-1203. doi: 10.1109/TVCG.2017.2657766 PMDI:28129180

[15] Cai, W., He, F., Lv, X., Cheng, Y. A Semi-Transparent Selective Undo Algorithm for Multi-user Collaborative Editors. Frontiers of Computer Science. 2021;15(5):1-17. doi: 10.1007/s11704-020-9518-x. 2021.

[16] Xu C, Lei X, Gao Z, et al. Direct delineation of myocardial infarction without contrast agents using a joint motion feature learning architecture[J]. Medical Image Analysis. 2018; S1361841518306960- .doi:10.1016/j.media.2018.09.001

[17] Caselles, V., Kimmel, R., Sapiro, G. Geodesic active contours, International Journal of Computer Vision. 1997; 22(1). doi: 10.1109/34.841758 

[18] Li, C., Xu, C., Gui, C., M.D. Fox. Distance regularized level set evolution and its application to image segmentation. IEEE Transactions on Image Processing. 2011; 19(12). doi: 10.1109/TIP.2010.2069690

[19] Liu, C., Liu, W., Xing, W. An improved edge-based level set method combining local regional fitting information for noisy image segmentation. Signal Processing. 2017; 130(1): 12-21. doi: 10.1016/j.sigpro.2016.06.013 

[20] Xing, W., Liu, C., Wei B. An improved edge-based level set method combining local regional fitting information for noisy image segmentation. Signal Processing: The Official Publication of the European Association for Signal Processing (EURASIP). 2017; 130: 12-21. doi: 10.1016/j.sigpro.2016.06.013 

[21] Khadidos, A., Sanchez, V., Li, C. T. Weighted level set evolution based on local edge features for medical image segmentation. IEEE Transactions on Image Processing. 2017; 26(4), 1979-1991. doi: 10.1109/TIP.2017.2666042

[22] Chan, T., Vese, L. Active contours without edges, IEEE Transaction on Image Processing. 2011; 10(2). doi:10.1109/83.902291 

[23] Li, C., Kao, C.Y., Gore, J. C., Ding, Z. Minimization of region-scalable fitting energy for image segmentation. IEEE Trans Image Process: a publication of the IEEE Signal Processing Society. 2008; 17(10): 1940–1949. doi: 10.1109/TIP.2008.2002304

[24] Liu, S., Peng, Y. A local region-based Chan–Vese model for image segmentation. Pattern Recognition. 2012; 45(7): 2769-2779. doi:10.1016/j.patcog.2011.11.019 

[25] Wang, X. F., Huang, D. S., Xu H. An efficient local Chan–Vese model for image segmentation. Pattern Recognition. 2010; 43(3): 603-618. doi: 10.1016/j.patcog.2009.08.002 

[26] Cheng L, Liu W, Xing W. An improved edge-based level set method combining local regional fitting information for noisy image segmentation. Signal Processing. 2017; 130(Jan.):12-21. doi: 10.1016/j.sigpro.2016.06.013 

[27] Zhang, K., Zhang, L., Song, H., Zhou, W. Active contours with selective local or global segmentation: a new formulation and level set method. Image and Vision Computing. 2010; 28(4), 668-676. doi: 10.1016/j.imavis.2009.10.009 

[28] Singh P. A novel hybrid time series forecasting model based on neutrosophic-PSO approach. International Journal of Machine Learning and Cybernetics. 2020; 11(10). doi: 10.1007/s13042-020-01064-z 

[29] Zhang D, You X, Liu S, et al. Multi-Colony Ant Colony Optimization Based on Generalized Jaccard Similarity Recommendation Strategy. IEEE Access. 2019; 7:157303-157317. doi: 10.1109/ACCESS.2019.2949860 

[30] Wang, H., Du, Y., Yi, J., Sun, Y., Liang, F. A new method for measuring topological structure similarity between complex trajectories. IEEE Transactions on Knowledge & Data Engineering. 2019; 31(10) :1836-1848. doi: 10.1109/TKDE.2018.2872523 

[31] Pan, Y, He, F., Yu, H. A novel Enhanced Collaborative Autoencoder with knowledge distillation for top-N recommender systems. Neurocomputing. 2019; 332:137-148. doi: 10.1016/j.neucom.2018.12.025 .

[32] Ronneberger O, Fischer P, Brox T. U-Net: Convolutional Networks for Biomedical Image Segmentation. Springer, Cham. 2015; doi:

10.1007/978-3-662-54345-0_3 

[33] Minaee S, YY Boykov, Porikli F, et al. Image Segmentation Using Deep Learning: A Survey. IEEE Transactions on Software Engineering. 2021; (99). doi:10.1109/TPAMI.2021.3059968

[34] Sultana F, Sufian A, Dutta P. Evolution of Image Segmentation using Deep Convolutional Neural Network: A Survey. Knowledge-Based Systems. 2020; s 201–202.doi: 10.1016/j.knosys.2020.106062

[35] Wang, B., Yuan, X., Gao, X., Li, X., Tao, D. A hybrid level set with semantic shape constraint for object segmentation. IEEE Transactions on Cybernetics. 2019; 49(5), 1558-1569. doi: 10.1109/TCYB.2018.2799999 

[36] Yan, Q., Yang, L., Liang, C., etc. Geometrically based linear iterative clustering for quantitative feature correspondence. Computer Graphics Forum. 2016; 35(7). Doi: 10.1111/cgf.12998 .

[37] Zhu, Z., Harowicz, M. R., Zhang, J., Saha, A., Mazurowski, M. A. Deep learning-based features of breast MRI for prediction of occult invasive disease following a diagnosis of ductal carcinoma in situ: preliminary data. Computer-aided Diagnosis. 2018; doi: 10.1117/12.2295470 

[38] Prasath V, Dang N, Hung N Q, et al. Multiscale Gradient Maps Augmented Fisher Information-Based Image Edge Detection. IEEE Access. 2020; 141104-141110, doi: 10.1109/ACCESS.2020.3013888 

[39] Hong, C., Yu, J., Zhang, J., Jin, X., Lee, K. H. Multimodal face-pose estimation with multitask manifold deep learning. IEEE transactions on industrial informatics. 2019; (15): 3952-3961, doi: 10.1109/TII.2018.2884211

8- The paper should be in the Plos one format not IEEE Access.

Answer:

 Thank you for your advice and patiently reviewing of our manuscript! we are very sorry for the irregular format, In the revised version, we have typeset in strict accordance with the standard format of Plos One.

Thank you again for carefully and patiently reviewing of our manuscript!

9- I think the authors should spend some time in revision justifying why this work is worthy of publication.

Answer:

 Thank you for your advice and patiently reviewing of our manuscript! In the revised version, we made a lot of changes to highlight our work.

First, in Section of “Introduction”, by citing preliminary work, we describe the new idea of the proposed the model, which is used to make up for the shortcomings of the existing methods.

Second, in Section of “Previous works”, Two subsections are added to the first subsection (CV model) and the fourth subsection (LRFI model), which are mainly used for comparison of segmentation models of the same type.

Third, in Section of “The proposed model”, we elaborated on the construction of the new model and the execution steps of the algorithm, and analyzed in detail the advantages of this model compared with other classic models.

Fourth, In Section of “Experimental results”, we have analyzed the experimental results in detail to verify the advantages of this model over other models.

Reviewer #2: This manuscript proposes a level-set-based method for image segmentation. The manuscript is clear, straightforward, easy to follow. However, I would like to draw the author's attention to the following major concerns:

Thanks very much for kind comment! 

Accordingly, based on your following detail comments, we improve the manuscript as much as possible in revised version one by one as follows.

1)Overall, the motivation is not introduced well, where the challenges should be described before the contributions. I recommend the authors to employ certain intuitive examples to elaborate the novelties of the proposed work.

Answer:

Thank you for your valuable comments! Your comments are very instructive and helpful for our future work. In revised version, we elaborate on our work to highlight the innovation of the model, which is mainly reflected in Sections of “Introduction”, “Previous works”, “The proposed Model”, and “Experimental results”.

The detail changes in revised version are as follows:

In the latter of Section "Introduction", we have further expanded the content and introduced our research model based on the shortcomings of the previous work.

In this paper, we propose a weighted region-based level set method to segment images with intensity inhomogeneity. In the model, first, a new weighted pressure force (WPF) is proposed to adaptively modulate the contractility of the balloon forces inside and outside of the closed contour by restricting the unevenness of the intensity mean value inside and outside the closed curve; second, a faster and smoother regularization term is utilized to ensure the stability of the curve evolution during the process of solving; finally, the WPF is integrated into the level set framework to improve the accuracy of segmentation images with intensity inhomogeneity.

In the section of “Previous works”, we added a new subsection to elaborate the idea of the proposed segmentation model by comparing with other standard models.

Summary

In summary, the four segmentation models based on the level set method have made great contributions to image segmentation with intensity inhomogeneity. As a typical segmentation model based on the level set, the CV model has the merit of high convergence speed if segmenting two phase images with intensity homogeneity. The DRLSE model normalizes the movement of the curve during the curve evolution process and improves the accuracy of the numerical solution. However, this model has poor performance in segmenting weak boundaries and high noise images. In the LVC model, the SPF is used to control the direction of the curve evolution which further improves the speed of iteration. The model improves the efficiency of the segmentation algorithm. Although this model is more efficient than the DRLSE model, this model loses its advantages when the image to be segmented has high intensity inhomogeneity and is less efficient when segmenting severe high-noise images. The LRFI model improves the efficiency of segmenting noisy images; however, it does not perform well in segmenting fuzzy boundary images.

Actually, the above segmentation models have their own advantages and disadvantages. How to make full use of the local information of an image is the focus of improving the accuracy of the segmentation model. In the proposed model, we utilize the statistical information of the local region to construct a weighted pressure force function, which adaptively shrinks and expands the closed contour. In addition, an enhanced distance regularized level set method is proposed to improve the speed of the segmentation algorithm, and to avoid the process of reinitialization.

In the section of “The proposed model”, we have expanded the content to better understand the ideas of the model in this manuscript.

The proposed model

Through the analysis of the above model, two core issues need to be solved: first, how to freely shrink and expand the closed contour by using the local information of an image; second, how to improve the accuracy of the segmentation model solution. Through in-depth research on the energy-based segmentation model in the previous work, this paper proposes a novel segmentation model based on local energy. Specifically, in this section, we present the details of the proposed weighted region-based ACM, which is based on the techniques of local statistical theory, curve evolution and the level set method…….

……

Advantage of the EDRLSM model over the other models

The advantage of the proposed model lies in the WPF function we constructed, which not only suppresses the influence of noise, thereby improving the accuracy of segmentation, but also increases the speed of the segmentation algorithm. 

 Compared with the classic CV model, the EDRLSM utilizes local information of the image to construct the WPF, and the proposed model establishes an energy function based on the local intensity information of the image, which greatly improves the accuracy of image segmentation with intensity inhomogeneity…….

In the section of “Experimental results”, We thoroughly analyzed the segmentation results of different types of images with intensity inhomogeneity to verify the advantages of the new model compared to other models.

Experimental results

This section shows experiments to demonstrate the effectiveness of the proposed model for both synthetic and real images. The proposed model is compared with the state-of-the art segmentation model based on level set methods to validate the effectiveness and robustness of our model. The proposed model is implemented in MATLAB R2018b on a 2.3 GHz Intel and 8.0GB RAM computer. 

……

Qualitative evaluation

 In the first experiment, we compare the proposed model with the LVC, GAC and DRLSE models in segmenting a synthetic image with noise. As shown in Figure 1, the image has the characteristics of fuzzy boundaries and multiple sharp corners, which brings greater difficulty to segmentation. The GAC, as a typical representative of energy-based models, has made a certain contribution to image segmentation. However, when the edges of the image are blurry or sharp corners, the segmentation accuracy of GAC model is significantly reduced. The regularization idea in The DRLSE model better standardizes the iteration of the curve and improves the accuracy of the solution, but the segmentation accuracy of the fuzzy boundary image is not high…….

……

Compared with the other models, the LVC model is the best in terms of segmentation speed, and the segmentation results are better. However, the generalization ability of this model is weak. For example, the segmentation effect of the model for the second image is poorer than that of the proposed model. Although the proposed model is not optimal in terms of segmentation speed, in terms of segmentation results, the segmentation effect of the model is best as shown in Figure 2. In the proposed model, the statistical information inside and outside the closed curve contour is utilized to construct the weighted pressure force, which greatly improves the accuracy of image segmentation with intensity inhomogeneity. Therefore, by comparing multiple models, the proposed model performs better in the accuracy of segmenting medical images.

Brain tissue segmentation has always been a research hotspot and difficulty in medical image segmentation. As shown in Figure 3, this experiment is mainly used to compare the accuracy of segmented brain tissue images. The proposed model makes full use of the local information of the image to construct the model and obtains a good segmentation effect. Specifically, Experiment 3 applied segmentation models to real medical images from slices of cerebral tissue taken via MRI…….

……

2)The paper does not explain clearly its advantages with respect to recent deep-learning literature: it is not clear what is the novelty and contributions of the proposed work: does it propose a new method? Or does the novelty only consist in the application?

Answer:

Thank you for your valuable comments! Your question is very good, and we have elaborated it in the section of “Introduction”. We proposed a new method which is based on active contour model. 

The detail changes in revised version are as follows:

In the second paragraph of section of “introduction”, we added a detailed explanation to expound clearly its advantages with respect to recent deep-learning literature.

In recent years, deep learning, such as convolutional neural networks (CNNs) are widely used to segment images and is have achieved excellent results in the medical image segmentation. However, these methods require considerable training dataset and have a complex structure and low execution efficiency. Although some deep segmentation models that support small sample data have recently appeared, these models often have over-segmentation or cross-learning problem. Therefore, Therefore, model-based image segmentation methods still have space for research, and these methods have higher time and space efficiency.

3) What is the motivation of the proposed work? Research gaps, objectives of the proposed work should be clearly justified. The authors should consider more recent research done in the field of their study

Answer:

 Thank you for your valuable comments! Your comments are very instructive and helpful for our future work.

First, we added two sections to the previous works section to introduce two recent classic segmentation models. 

Second, we added a summary to the previous works section to summarize the previous work and explain the research gap.

The detail changes in revised version are as follows:

In the “Previous works” section, two subsections are added to the first subsection (CV model) and the fourth subsection (LRFI model), respectively.

Previous works

CV model

The CV model, as a classic typical energy-based segmentation model, is proposed to segment two-phase images, based on the assumption that the target is segmented and the background is intensity inhomogeneous. Let an image I(x) on the image domain �, the energy function is defined as:

 ……

 According to the method of variational method, the evolution equation of the curve is expressed as follows:

 ……

 The CV model can well segment images with intensity inhomogeneity, however, it has more sensitivity to initialization information and low efficiency while segmenting images with high noise and severe intensity inhomogeneity.

……

LRFI model

Liu et al. utilized local regional fitting information to propose an improved level set method, which can differentiate the noise and boundary points of the image to be segmented. In this model, two innovations are proposed: first, a controllable velocity coefficient was proposed to accelerate the curve convergence speed, and second, a new edge stop function was constructed to enhance the performance of the segmentation model. The velocity function was shown as:

 ……

where fin(x) and f(x) are local regional fitting means of image pixels inside and outside of the closed contour, respectively, and � and k are two positive coefficients.

And, the edge stop function (ESF) was defined as:

……

According to the principle of the variational method, the curve iteration function of the model is described as follows:

 ……

Compared with the DRLSE model, the function v(x) of the LRFI model can better make the closed curve converge along the object boundaries, and the ESF improves the accuracy of the numerical calculation. The LRFI model demonstrates good performance in segmenting noisy images.

In the last subsection, summary is added to summarize the previous work and explain the research gap.

Summary

In summary, the four segmentation models based on level set method have made great contributions to images segmentation with intensity inhomogeneity. As a typical segmentation model based on level set, The CV model has the merit of high convergence speed if segmenting two phase images with intensity homogeneity. The DRLSE model normalizes the movement of the curve during the curve evolution process and improves the accuracy of the numerical solution. However, this model has poor performance in segmenting weak boundaries and high noise images. In the LVC model, the SPF is used to control the direction of the curve evolution which further improves the speed of iteration. The model improves the efficiency of the segmentation algorithm. Although this model is more efficient than the DRLSE model, this model loses its advantages when the image to be segmented has high intensity inhomogeneity and is less efficient when segmenting severe high-noise images. The LRFI model improves the efficiency of segmenting noisy images, however, it does not perform well in segmenting fuzzy boundary images.

Actually, the above segmentation models have their own advantages and disadvantages. How to make full use of the local information of the image is the focus of improving the accuracy of the segmentation model. In the proposed model, we utilize the statistical information of the local region to construct a weighted pressure force function, which adaptively shrink and expand the closed contour. In addition, an enhanced distance regularized level set method is proposed to improve the speed of segmentation algorithm, and to avoid the process of reinitialization.

4) Authors should add more details about the implementation of the code to perform the analysis and the dateset involved in this task.

Answer:

 Thank you for your valuable comments! Your comments are very instructive and helpful for our future work.

 First, In the section of “Experimental results”, a more detailed description of the experimental environment was added.

 Second, we conducted a more comprehensive analysis of the experimental comparison results.

The detail changes in revised version are as follows:

In the first paragraph of “Experimental results”, a more detailed description is added.

Experimental results

This section shows experiments to demonstrate the effectiveness of the proposed model for both synthetic and real images. The proposed model is compared with the state-of-the art segmentation model based on level set methods to validate the effectiveness and robustness of our model. The proposed model is implemented in MATLAB R2018b on a 2.3 GHz Intel and 8.0GB RAM computer. 

A more comprehensive analysis of the experimental comparison results is involved in this task.

Qualitative evaluation

In the first experiment, we compare the proposed model with the LVC, GAC and DRLSE models in segmenting a synthetic image with noise. As shown in Figure 1, the image has the characteristics of fuzzy boundaries and multiple sharp corners, which brings greater difficulty to segmentation. The GAC, as a typical representative of energy-based models, has made a certain contribution to image segmentation. However, when the edges of the image are blurry or sharp corners, the segmentation accuracy of GAC model is significantly reduced. The regularization idea in The DRLSE model better standardizes the iteration of the curve and improves the accuracy of the solution, but the segmentation accuracy of the fuzzy boundary image is not high. As shown in Figure 1, compared with the GAC and DRLSE models, the segmentation accuracy of this model is higher at the fuzzy angles of the image. As a representative edge-based LSM, the GAC and DRLSE models perform poorly, mainly because the synthetic image is affected by noise and blurred boundaries.

……

As shown in Figure 2,……Compared with the other models, the LVC model is the best in terms of segmentation speed, and the segmentation results are better. However, the generalization ability of this model is weak. For example, the segmentation effect of the model for the second image is poorer than that of the proposed model. Although the proposed model is not optimal in terms of segmentation speed, in terms of segmentation results, the segmentation effect of the model is best as shown in Figure 2. In the proposed model, the statistical information inside and outside the closed curve contour is utilized to construct the weighted pressure force, which greatly improves the accuracy of image segmentation with intensity inhomogeneity. Therefore, by comparing multiple models, the proposed model performs better in the accuracy of segmenting medical images…….

Brain tissue segmentation has always been a research hotspot and difficulty in medical image segmentation. As shown in Figure 3, this experiment is mainly used to compare the accuracy of segmented brain tissue images. The proposed model makes full use of the local information of the image to construct the model and obtains a good segmentation effect. Specifically, ……

5) Considering that deep-learning-based methods (Such as Mask RCNN, Unet) have shown impressive performance in image segmentation, the advantages and disadvantages of the proposed method should be discussed more clearly, and presented in the experiment.

Answer:

 Thank you for your valuable comments! Your comments are very instructive and helpful for our future work. 

 Deep-learning-based methods have shown impressive performance in image segmentation, since our segmentation model is for small sample data, the proposed model is based on the energy-based model. We give an explanation in the Introduction section. Of course, the learning-based method has its advantages, which is the direction of our future research.

6) The authors should do a more thorough literature survey. Just to name a few:

- Ronneberger, O., Fischer, P., & Brox, T. (2015, October). U-net: Convolutional networks for biomedical image segmentation. In International Conference on Medical image computing and computer-assisted intervention (pp. 234-241). Springer, Cham.

- Xu, C., Xu, L., Gao, Z., Zhao, S., Zhang, H., Zhang, Y., ... & Li, S. (2018). Direct delineation of myocardial infarction without contrast agents using a joint motion feature learning architecture. Medical image analysis, 50, 82-94.

- Minaee, S., Boykov, Y. Y., Porikli, F., Plaza, A. J., Kehtarnavaz, N., & Terzopoulos, D. (2021). Image segmentation using deep learning: A survey. IEEE Transactions on Pattern Analysis and Machine Intelligence.

- Sultana, F., Sufian, A., & Dutta, P. (2020). Evolution of image segmentation using deep convolutional neural network: a survey. Knowledge-Based Systems, 201, 106062.

Answer:

 Thank you very much for your valuable comments which improves the quality of this manuscript.

 First, we have added relevant references as needed;

 Second, we add relevant references in accordance with the standard format required by PLOS ONE for reference documents, as shown in references [16], [32],[33],[34].

 The detail changes in revised version are as follows: 

 [16] Xu C, Lei X, Gao Z, et al. Direct delineation of myocardial infarction without contrast agents using a joint motion feature learning architecture[J]. Medical Image Analysis. 2018; S1361841518306960-. doi:10.1016/j.media.2018.09.001

……

[32] Ronneberger O, Fischer P, Brox T. U-Net: Convolutional Networks for Biomedical Image Segmentation. Springer, Cham. 2015; doi: 10.1007/978-3-662-54345-0_3 

[33] Minaee S, YY Boykov, Porikli F, et al. Image Segmentation Using Deep Learning: A Survey. IEEE Transactions on Software Engineering, 2021, (99). doi:10.1109/TPAMI.2021.3059968

[34] Sultana F, Sufian A, Dutta P. Evolution of Image Segmentation using Deep Convolutional Neural Network: A Survey[J]. Knowledge-Based Systems, 2020, s 201–202.doi:10.1016/j.knosys.2020.106062

7) There are some grammar errors and typos. I suggest the authors make an solid, overall proofreading.

Answer:

Thank you very much for your valuable comments, which is a great help not only to improve the quality of this manuscript but also to instruct our research in future! 

After we carefully revised the manuscript based on expert reviews, then the manuscript was edited for proper English language, grammar, punctuation, spelling, and overall style by one or more of the highly qualified native English-speaking editors at AJE.

Thank you very much for your valuable comments, again!

---

## [Decision Letter · Decision Letter 1]

28 Jul 2021

A weighted region-based level set method for image segmentation with intensity inhomogeneity

PONE-D-21-13529R1

Dear Dr. Yu,

We’re pleased to inform you that your manuscript has been judged scientifically suitable for publication and will be formally accepted for publication once it meets all outstanding technical requirements.

Kind regards,

Zhifan Gao

Academic Editor

PLOS ONE

Additional Editor Comments (optional):

Reviewers' comments:

Reviewer's Responses to Questions

**Comments to the Author**

1. If the authors have adequately addressed your comments raised in a previous round of review and you feel that this manuscript is now acceptable for publication, you may indicate that here to bypass the “Comments to the Author” section, enter your conflict of interest statement in the “Confidential to Editor” section, and submit your "Accept" recommendation.

Reviewer #1: All comments have been addressed

Reviewer #2: All comments have been addressed

2. Is the manuscript technically sound, and do the data support the conclusions?

Reviewer #1: Yes

Reviewer #2: Yes

3. Has the statistical analysis been performed appropriately and rigorously? 

Reviewer #1: Yes

Reviewer #2: Yes

4. Have the authors made all data underlying the findings in their manuscript fully available?

Reviewer #1: Yes

Reviewer #2: Yes

5. Is the manuscript presented in an intelligible fashion and written in standard English?

Reviewer #1: Yes

Reviewer #2: Yes

6. Review Comments to the Author

Reviewer #1: In revised version, the authors have been addressed all comments regarding technical, language and format.

Thank you

Reviewer #2: (No Response)

7. PLOS authors have the option to publish the peer review history of their article (what does this mean?). If published, this will include your full peer review and any attached files.

Reviewer #1: No

Reviewer #2: No

---

## [Editor Report · Acceptance letter]

9 Aug 2021

PONE-D-21-13529R1 

A weighted region-based level set method for image segmentation with intensity inhomogeneity 

Dear Dr. Yu:

I'm pleased to inform you that your manuscript has been deemed suitable for publication in PLOS ONE. Congratulations! Your manuscript is now with our production department. 

Kind regards, 

on behalf of

Dr. Zhifan Gao 

Academic Editor

PLOS ONE